# Predictors of depression in asthma patients in an Indian referral hospital

**Anushka Kataria**[1], **Jefferson Daniel**[1*], **Barney Isaac**[1], **Balamugesh Thangakunam**[1], **Jesson Paulson Illimoottil**[2], **Reka Karuppusami**[3], **Devasahayam J. Christopher**[1]

**1** Department of Pulmary Medicine, Christian Medical College, Vellore, Tamil Nadu, India, **2** Department of Psychiatry, Unit III, Christian Medical College, Vellore, Tamil Nadu, India, **3** Department of Biostatistics, Christian Medical College, Vellore, Tamil Nadu, India

* jefferson.daniel@cmcvellore.ac.in

## Abstract

### Background

Asthma has been shown to have a significant association with the diagnoses of mental health disorders, including depression; however, information from India is scarce.

### Objectives

The study aims to estimate the prevalence of depression in asthma patients and identify associated risk factors.

### Methods

A total of 248 participants aged 18 and above who presented to the pulmonary medicine outpatient clinic of the Christian Medical College, Vellore, Tamil Nadu, India, were recruited. Socio-demographic and clinical details, including treatment compliance were collected, and inhaler technique was assessed. Asthma control was measured by the Asthma Control Test (ACT) and the quality of life by Asthma Quality of Life Questionnaire (AQLQ). The Patient Health Questionnaire-9 (PHQ-9) was used to screen for depression. Logistic regression was used to determine the risk factors for depression.

### Results

Out of the 248 participants, 24.6% screened positive for depression. Asthma control was worse (ACT 19.07±4.87 vs 21.93±3.66; p<0.001) and quality of life was poorer (AQLQ 5.53±0.72 vs 6.06±0.63; p<0.001) in those with depression. There was a low negative correlation ($\rho=-0.332$, P<0.001) between ACT score and PHQ-9 score and a moderate negative correlation ($\rho=-0.451$, P<0.001) between AQLQ score and PHQ-9 score. Univariate logistic regression analysis showed six independent factors associated with depression: female gender (OR 1.87), presence of one or more

**Data availability statement:** All relevant data are within the paper and its Supporting Information files.

**Funding:** This study was supported by the intramural Fluid Research Grant, provided by the Office of Research, Christian Medical College, Vellore (IRB Minute no. 15251). https://www.cmch-vellore.edu/research-office/ The funders had no role in study design, data collection and analysis, decision to publish, or preparation of the manuscript.

**Competing interests:** The authors have declared that no competing interests exist.

comorbidities (OR 2.10), incorrect inhaler technique (OR 3.59), poor compliance with inhaled corticosteroids (ICS) (OR 2.30), poorly controlled asthma (ACT) (OR 2.92) and lower AQLQ score (OR = 0.32). Multivariate logistic regression analysis revealed that the AQLQ was the only independent risk factor for depression (OR 0.27, P = 0.001).

## Conclusion

Adult asthma patients in this Indian cohort showed a high prevalence of depression. The study highlights key risk factors for depression in asthma and reiterates the need to screen asthma patients for depression.

## Introduction

Asthma is a common chronic respiratory disease with an estimated total burden of 34.3 million in India, accounting for about 13% of the global burden [1]. While it has a relatively low mortality rate, it is associated with a high morbidity, globally ranking 16th among the leading causes of years lived with disability and 28th among the leading causes of disease burden, as measured by disability-adjusted life years [2]. In addition to causing a decline in physical health, asthma is also associated with mental health disorders. [3–5]. Several studies done worldwide have found a higher prevalence of depression in both children and adults with asthma. [5–8]. While some studies point towards a bidirectional relationship between asthma and depression [9], others have shown that only depression is associated with an increased risk of developing adult-onset asthma and not vice-versa [10,11].

Much research has recently focused on exploring the possible risk factors associated with depression in those with asthma. It has been postulated that varying levels of specific cytokines may have an essential role in arousing and remitting both asthma and depression, suggesting that an inflammatory response could be a common pathway between the two conditions [12].

Given the substantial asthma morbidity, and its impact on mental health, the prevalence of depression was important to ascertain. There is a dearth of good quality data on the prevalence of depression in Indian asthma patients. Therefore, to address this knowledge gap, we designed this study with the aim of assessing the prevalence of depression, and identifying the associated risk factors.

## Methodology

This was a cross-sectional study conducted between 31.03.2023 and 18.07.2023, after obtaining approval from the Institutional Review Board (IRB Minute no. 15251). Adults with asthma presenting to the Pulmonary Medicine out-patient clinic at a tertiary center in southern India, were invited to participate in the study. Written informed consent was obtained from the participants prior to enrollment. Consecutive patients aged 18 years or more, presenting to the Pulmonary Medicine outpatient clinic, were screened. Those with a physician diagnosis of asthma were considered for recruitment by simple random sampling on each outpatient day, ensuring that 10–15 participants were recruited per day. The investigators screened patients to ensure

that the recruited patients fulfilled diagnostic criteria for asthma based on Global Initiative for Asthma (GINA) guidelines (2022) [13]. The GINA guidelines stipulate diagnosing asthma when there is either a history of variable respiratory symptoms (wheeze, shortness of breath, chest tightness and cough), associated with documented expiratory airflow limitation (FEV1 is reduced, FEV1/FVC is lesser than 0.7 for adults) or post-bronchodilator reversibility (increase in FEV1 of >12% and >200 mL after administration of bronchodilator). Those with symptoms for less than a year were excluded. Assuming a 65% prevalence of depression in asthma patients, based on an earlier Indian study by Misra et al [14], with an alpha error of 5% and a precision of 6%, a sample size of 243 was needed.

Relevant socio-demographic information was recorded. Historical information obtained included duration of symptoms, treatment history, comorbidities, history of smoking, alcohol and recreational drug use, and personal or family history of psychiatric illness. The Asthma Control Test (ACT) was used to assess asthma control [15]. The ACT assesses the frequency of shortness of breath and general asthma symptoms, use of rescue medications, the effect of asthma on daily functioning, and overall self-assessment of asthma control, with 4-week recall. Items are graded on a 5-point scale (for symptoms and activities: 1 = all the time to 5 = not at all, and for asthma control rating: 1 = not controlled at all to 5 = completely controlled) and the total scores range was from 5 (poor control of asthma) to 25 (complete control of asthma), with higher scores reflecting greater asthma control. An ACT score of 19 or higher is considered indicative of well-controlled asthma. Relevant physical examination findings were also recorded.

Spirometry was performed and interpreted according to American Thoracic Society (ATS) guidelines, for acceptability and reproducibility criteria [16]. Blood tests ordered included eosinophil count, and an absolute eosinophil count (AEC) >300 cells/µL was considered to indicate peripheral eosinophilia. Fractional exhaled nitric oxide (FeNO) was measured with the Fenom Pro FeNO monitoring system, and a FeNO value > 25 ppb was considered airway eosinophilia. For the purpose of this study, eosinophilic asthma was defined as the presence of either peripheral or airway eosinophilia or both.

The Asthma Quality of Life Questionnaire (AQLQ) was used to assess the quality of life [17]. It is a disease-specific health-related quality of life instrument that assesses both physical and emotional impact of disease with a 2-week recall. It includes a total of 32 items divided into 4 domains: symptoms (11 items), activity limitation (12 items, 5 of which are individualized), emotional function (5 items), and environmental exposure (4 items). Items are graded on a 7-point Likert scale (7 = not impaired at all – 1 = severely impaired), with higher scores indicating better quality of life.

Participants were screened for depression using the Patient Health Questionnaire-9 (PHQ-9), a self-reported tool, which is a multipurpose instrument for screening, monitoring and measuring the severity of depression [18–20]. PHQ-9 was selected for its brevity and ease of administration. It scores each of the nine DSM-IV criteria as "0" (not at all) to "3" (nearly every day). A total PHQ-9 score of ≥ 5 was used to diagnose depression. The grading of severity was based on score: 5–9 – mild, 10–14 – moderate, 15–19 – moderately severe, and 20–27 – severe depression. Study data was collected and managed using REDCap electronic data capture tools hosted by the institution.

## Study variables and statistical tests

Data was analyzed using SPSS V 21{License Number (Customer ID): 200699; Vendor: SPSS South Asia Pvt. Ltd. Bangalore}. Continuous variables were expressed as mean and standard deviation, or median and interquartile range, based on the normality. Categorical variables were expressed as frequencies and percentages. In the univariate analysis, an independent t-test or Mann-Whitney U test was used for continuous variables, and a Chi-square test was used for categorical variables. The odds ratio (OR) with 95% confidence interval (CI) was calculated to determine the strength of association. Pearson correlation coefficient and Spearman's Rank correlation coefficient were used to study the correlation between variables. Statistical analyses were performed using available data. Participants with missing values for a specific variable were excluded only from that particular analysis. Differences were considered statistically significant at the two-sided p-value < 0.05 level, Multivariate logistic regression analysis was performed to adjust for the confounding factors, and an adjusted odds ratio was obtained to ascertain independent predictors of depression.

 

## Results

Fig 1 shows the CONSORT flow diagram of the study. A total of 842 patients with a prior diagnosis of asthma were considered for recruitment into the study. Simple random sampling was used to recruit 264 patients, of whom 248 were considered for the final analysis, after undergoing study-related assessments. The participants had a median duration of asthma symptoms of 7.5 years (interquartile range [IQR] 3–15 years) and a duration since diagnosis of asthma of 36 months (IQR 12–120 months). Among these, 21.4% were asthma treatment naive.

Tables 1 and 2 show a clinico-socio-demographic profile of participants in relation to the presence of depression. The mean age at enrollment was 44.6 years (SD 12.97), 52.4% were female and 83.5% were married. The mean body mass index (BMI) was 25.6 kg/m$^2$ and 16.5% had a BMI ≥ 30 kg/m$^2$. As assessed by the Kuppuswamy scale (2022), 56% belonged to the lower middle, upper middle or upper class, while the remaining 44% belonged to the upper lower or lower lower class. 52.4% lived in rural areas. In all, 14.9% reported a history of smoking, 6.5% were current smokers, 8.5% reported alcohol consumption and none reported recreational drug use. Overall, 61.3% of the participants reported having one or more comorbidities. 3.2% had a prior history of psychiatric illness, 75% of which was depression, and 0.8% were currently on antidepressants. Overall, 4% reported a family history of psychiatric illnesses.

Among those on inhaled corticosteroids (ICS), 93.7% demonstrated correct inhaler technique and 66.2% were compliant with inhalers. The median duration of ICS use was 48 months (IQR 12–96). In all, 25% had used monteleukast in the past, with a median duration of use of 9 months (IQR 3–36), and 4.8% had been prescribed oral steroids in the past.

The participants had a mean pre-bronchodilator FEV1 of 70.98% (SD 20.91). The median AEC and FeNO was 264/μL (IQR 140–430) and 19 ppb (IQR 12–35), respectively. 60.5% of the participants fulfilled criteria for eosinophilic asthma – 43.1% had peripheral eosinophilia (AEC > 300), 37.1% had airway eosinophilia (FeNO value >25 ppb) and 19.8% had

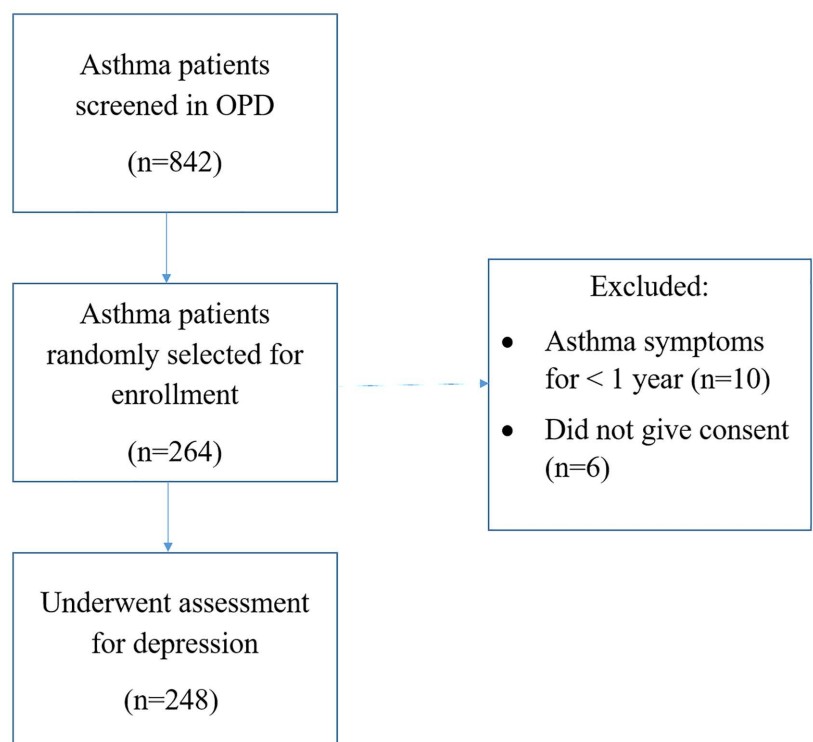

**Fig 1. CONSORT flow diagram of the cross-sectional study.** OPD, Out-patient department.

**Table 1. Demographics and clinico-social profile.**

| Sample Characteristics | Total participants [n(%)/ Mean(SD)/ Median(IQR)] (N = 248) | No depression [n(%)/ Mean(SD)/ Median(IQR)] (n = 187) | Depression [n(%)/ Mean(SD)/ Median(IQR)] (n = 61) | P-value |
|---|---|---|---|---|
| Gender, female | 130 (52.4%) | 91(70%) | 39(30%) | **0.038** |
| Age | 44.6 (12.97) | 44.88 (13.24) | 43.74 (12.15) | 0.55 |
| BMI | 25.6 (4.93) | 25.66 (4.77) | 25.45 (5.44) | 0.77 |
| Obese (BMI ≥ 30 kg/m$^2$) | 41(16.5%) | 31(75.6%) | 10 (24.4%) | 0.97 |
| SES – modified Kuppuswamy scale (2022) | | | | 0.26 |
| SES I, II and III | 139(56%) | 101(72.7%) | 38(27.3%) | |
| SES IV and V | 109(44%) | 86(78.9%) | 23(21.1%) | |
| Residence – Urban/urban slum | 118(47.6%) | 91(77.1%) | 27(22.9%) | 0.55 |
| Unmarried/divorced/widowed | 41(16.5%) | 30(73.2%) | 11(26.8%) | 0.72 |
| Past history of smoking | 37(14.9%) | 32(86.5%) | 5 (13.5%) | 0.09 |
| Active smoker | 16 (6.5%) | 14 (87.5%) | 2 (12.5%) | 0.37 |
| Smoking pack years (n = 37) | 2 (0.5-10) | 2 (0.5-10) | 5 (0.15-13) | 0.91 |
| History of alcohol consumption | 21(8.5%) | 17 (81%) | 4 (19%) | 0.54 |
| Comorbidities | 152(61.3%) | 107 (70.4%) | 45 (29.6%) | **0.021** |
| History of psychiatric illness | 8 (3.2%) | 4 (50%) | 4 (50%) | 0.10 |
| Family history of psychiatric illness | 10 (4%) | 6 (60%) | 4 (40%) | 0.27 |
| Duration of symptoms (in years) | 7.5 (3-15) | 8 (3-15) | 7 (3.5-15.5) | 0.85 |
| Time since diagnosis of asthma made (in months) | 36 (12-120) | 30 (8-120) | 60 (12-132) | 0.35 |
| Asthma severity based on pre-bronchodilator FEV1 (in percentage) | 70.98 (20.91) | 70.89 (20.54) | 71.24 (22.18) | 0.91 |
| Treatment naïve | 53 (21.4%) | 37 (69.8%) | 16 (30.2%) | 0.29 |
| Duration of ICS use (in months) (n = 195) | 48 (12-96) | 48 (12-99) | 36 (11.5-78) | 0.37 |
| Correct inhaler technique | 179 (93.7%) | 140 (78.2%) | 39 (21.8%) | 0.06 |
| Poor compliance with inhalers | 66 (33.8%) | 44 (66.7%) | 22 (33.3%) | **0.015** |
| Use of Monteleukast | 62 (25%) | 46 (74.2%) | 16 (25.8%) | 0.80 |
| Use of oral steroids | 12 (4.8%) | 10 (83.3%) | 2 (16.7%) | 0.74 |
| Eosinophilic asthma (peripheral and/or airway) | 150 (60.5%) | 112 (74.7%) | 38 (25.3%) | 0.74 |
| Eosinophilia (both, peripheral and airway) | 49 (19.8%) | 39 (79.6%) | 10 (20.4%) | 0.45 |
| Poorly controlled asthma (ACT ≤ 19) | 67(27.0%) | 40 (59.7%) | 27 (40.3%) | **<0.001** |

N = total sample size; n = subgroup size; SD = standard deviation; IQR = interquartile range.

BMI, body mass index; SES, socioeconomic status; FEV1, forced expiratory volume in 1 second; ICS, inhaled corticosteroid; AEC, absolute eosinophil count; FeNO, fractional exhaled nitric oxide; ACT, Asthma Control Test.

both peripheral and airway eosinophilia. The mean ACT score was 21.23(SD 4.17), and 73% had controlled asthma. The mean AQLQ score was 5.93(SD 0.69).

The PHQ-9 questionnaire revealed that 24.6% participants had depression (95% CI [19.4,30.4]). Of these, 18.1% had mild, 5.2% had moderate, 1.2% had moderately severe and none had severe depression. Rates were higher in females than males (30% vs 18.6%; P = 0.038), and in those with one or more comorbidities compared to those without (29.6% vs 16.7%; P = 0.021). Analyzed individually, the comorbidities (diabetes mellitus, hypertension, dyslipidemia, coronary artery disease, ischemic heart disease, hypothyroidism, chronic kidney disease, chronic liver disease, interstitial lung disease, bronchiectasis and rheumatoid arthritis) did not reveal statistically significant associations with depression. Depression was less common among those who were compliant with the use of ICS compared to those who reported non-compliance (17.8% vs. 33.3% P = 0.015), and in those with controlled asthma compared to those who were poorly controlled (18.8%

 

**Table 2. Lab parameters and scores.**

| Lab parameters/scores | Total participants [n(%)/ Mean(SD)/ Median(IQR)] (N=248) | No depression [n(%)/ Mean(SD)/ Median(IQR)] (n=187) | Depression [n(%)/ Mean(SD)/ Median(IQR)] (n=61) | P-value |
|---|---|---|---|---|
| AEC | 264.00 (140.0-430.0) | 247.00 (139.5-423.0) | 288.00 (146.0-457.5) | 0.59 |
| FeNO | 19 (12-35) | 19 (12-36) | 18 (12.5-34) | 0.98 |
| Peripheral eosinophilia (AEC>300/µL) | 107 (43.1%) | 79 (73.8%) | 28 (26.2%) | 0.62 |
| Airway eosinophilia (FeNO>25 ppb) | 92 (37.1%) | 72 (78.3%) | 20(21.7%) | 0.42 |
| ACT score | 21.23 (4.17) | 21.93 (3.66) | 19.07(4.87) | **<0.001** |
| AQLQ score | 5.93 (0.69) | 6.06 (0.63) | 5.53 (0.72) | **<0.001** |

N=total sample size; n=subgroup size; SD=standard deviation; IQR=interquartile range.

AEC, absolute eosinophil count; FeNO, fractional exhaled nitric oxide; ACT, Asthma Control Test; AQLQ, Asthma Quality of Life Questionnaire.

vs 40.3%; P<0.001). There was no difference between those with eosinophilic asthma compared to those with non-eosinophilic asthma (25.3% vs. 23.5%; P=0.62). Patients with depression had poorer asthma control and quality of life scores compared to those without [(ACT 19.07±4.87 vs 21.93±3.66; d=2.86; p<0.001) and (AQLQ 5.53±0.72 vs 6.06±0.63; d=0.53; p<0.001), respectively]. History of monteleukast use did not appear to be associated with depression.

Correlation analysis showed a weak, statistically significant negative correlation between ACT score and PHQ-9 score (ρ=−0.332, P<0.001) (Fig 2A), and a moderate, statistically significant negative correlation between AQLQ score and PHQ-9 score (ρ=−0.451, P<0.001) (Fig 2B). This suggested a correlation of depression with poor asthma control and worse quality of life. Correlation analysis between FeNO and ACT score showed a negligible, but statistically significant negative correlation (ρ=−0.135, P=0.034), suggesting a correlation between higher FeNO values and poor asthma control.

Logistic regression was used to analyze the relationship between selected clinical parameters, ACT, AQLQ and depression (Table 3). The univariate logistic regression analysis identified six independent factors associated with depression in

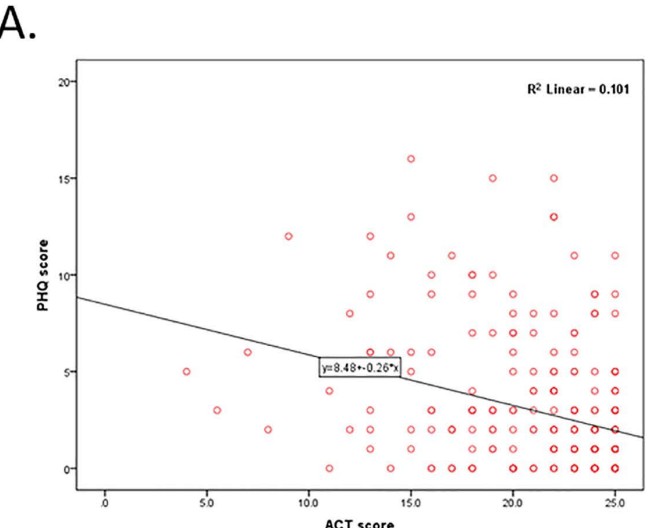
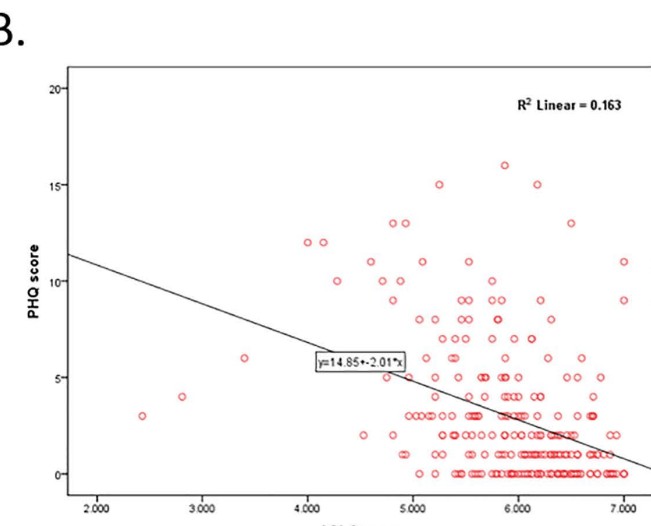

**Fig 2. Scatter plot graphs of the Spearman correlation analysis between depression and asthma control (A) and quality of life (B).** PHQ, Patient Health Questionnaire; ACT, Asthma Control Test; AQLQ, Asthma Quality of Life Questionnaire.

**Table 3. Logistic regression model: risk factors for depression in asthma patients.**

| Predictor variables | Univariate regression | | Multivariate regression[a] | |
|---|---|---|---|---|
| | Unadjusted OR [95% CI] | P-value | Adjusted OR [95% CI] | P-value |
| Female gender | 1.87 [1.03,3.39] | **0.040** | 1.89 [0.87,4.14] | 0.11 |
| Comorbidities | 2.10 [1.11,3.99] | **0.023** | 1.80 [0.81,3.96] | 0.15 |
| Inhaler technique incorrect | 3.59 [1.10,11.75] | **0.035** | 1.44 [0.33,6.19] | 0.63 |
| Non-compliance with ICS | 2.30 [1.16,4.56] | **0.016** | 1.69 [0.77,3.70] | 0.19 |
| ACT (Partly controlled/uncontrolled asthma) | 2.92 [1.58,5.40] | **0.001** | 0.97 [0.39,2.44] | 0.95 |
| AQLQ | 0.32 [0.20,0.52] | **<0.001** | 0.27 [0.12,0.57] | **0.001** |
| Non-eosinophilic asthma | 1.11 [0.61,2.00] | 0.739 | | |

OR = odds ratio; CI = confidence interval.

ICS, inhaled corticosteroid; ACT, Asthma Control Test; AQLQ, Asthma Quality of Life Questionnaire.

[a]Multivariate model included female gender, presence of comorbidities, incorrect inhaler technique, non-compliance with ICS, ACT score and AQLQ score.

asthma patients: female gender (OR 1.87), presence of one or more comorbidities (OR 2.10), incorrect inhaler technique (OR 3.59), non-compliance with ICS (OR 2.30), poorly controlled asthma (ACT) (OR 2.92) and lower AQLQ score (OR = 0.32). In the multivariate logistic regression analysis (Table 3), only AQLQ score remained a significant predictor of depression (OR 0.27, P = 0.001). The loss of significance for other variables suggested that their effects might have been confounded. Hence, multi-collinearity was evaluated using the variance inflation factor (VIF) (all values < 5), and confounding was examined using both statistical assessment and clinical judgement; no major confounding effects were identified, and odds ratios remained stable across models. Stepwise regression identified four variables as the best predictors, but their inclusion did not substantially change the odds ratios. Therefore, the full multivariate model with six variables was reported for completeness.

## Discussion

This study provided an estimate of the prevalence of depression in Indian patients. Nearly a quarter of the patients screened positive for depression, and a quarter of these had severe depression. These rates were higher than the 11−21% reported from other countries [7,21,22]. Higher rates of depression among Indian asthma patients was corroborated by another study done in eastern India, which reported an even higher prevalence (65%) among out-patients visiting a tertiary care hospital [14]. It is important to note that this Eastern Indian study used a different tool (Beck Depression Inventory (BDI) score) to evaluate depression, making precise comparison with our study difficult. The BDI is more detailed than the PHQ-9, and covers a wider range of symptoms. However, both the BDI and PHQ-9 have shown a close correlation in diagnosing depression [23–25]. The PHQ-9 can be completed within 3 minutes, and was thus selected for its feasibility for administration in our busy outpatient clinic.

Female gender, presence of one or more comorbidities, poor compliance with inhalers and poorly controlled asthma were associated with a higher prevalence of depression. The prevalence of depression was found to be significantly higher in females compared to males. This finding is supported by the 'social signal transduction theory of depression'. The theory describes neurophysiologic, molecular, and genomic mechanisms linking social and environmental adversity with biological processes that drive depression pathogenesis, maintenance, and recurrence [26]. Based on this theory, stress activates pro-inflammatory cytokines, which drive the depression pathogenesis. Fluctuations in ovarian hormones affect women's susceptibility to stress, alter brain structure and function, and regulate inflammatory processes and responsiveness [27]. Some studies found evidence of the influence of gender on the bidirectional relationship between depression and asthma, with one study showing that compared to male patients, female patients were more likely to

experience asthma exacerbations at low and moderate levels of depression. However, this disparity disappeared at high levels of depression [28]. A study done in the USA found that depression in asthma patients was associated with higher mortality, but after adjusting for comorbidities, depression remained an independent risk factor for mortality only in female patients [29].

Prevalence of depression was significantly higher in those with one or more comorbidities. Other studies reported a similar association [30,31]. An India study showed that patients with a single chronic medical condition were 3–4 times more likely to experience depression or anxiety compared to those without, and this increased to 6-fold for those with two or more medical conditions [32]. The increased prevalence of depression in those with asthma and other comorbidities could also be explained by the 'social signal transduction theory of depression', which suggests that several somatic conditions – such as asthma, rheumatoid arthritis, chronic pain, metabolic syndrome, cardiovascular disease, obesity, and neurodegeneration – activate the pro-inflammatory cytokines that drive the pathogenesis of depression.

Prevalence of depression was significantly lower in those who were compliant with ICS use in our cohort, which perhaps may be the result of better symptom and disease control. However, previous studies showed no consistent correlation between ICS use and depression [33]. The relationship between depression, medication adherence, and asthma control is likely complex and multidirectional. Depression and non-compliance may co-occur through shared underlying factors such as motivation or illness perception, though the temporal sequence cannot be determined from our cross-sectional data.

The prevalence of depression was significantly lower in those with controlled asthma compared to those with partially controlled or uncontrolled asthma. This finding is consistent with previous studies, which have shown an association between depression and poor asthma control [21,33–37]. While many studies found depression to be independently associated with poor asthma control and increased hospital visits [22,35,36,38–44], one study reported that the increased likelihood of depression among patients with asthma was not exclusively related to severe or poorly controlled asthma [45]. Another study found depression in asthma patients to be associated with a reduced bronchodilator response [46]. Moreover, some studies also suggested that treating comorbid depression in asthma patients could improve overall patient outcomes. Untreated depression in those with asthma had a higher prevalence of airway obstruction compared to asthma patients receiving anti-depressants [47]. One study demonstrated that after six months of guideline-driven standardized asthma treatment, both asthma outcomes and psychological outcomes (anxiety and depression) improved [40]. Brown et al reported that patients with severe asthma who had major depressive disorder showed significant improvement in their asthma control after 12 weeks of treatment with a selective serotonin reuptake inhibitor (SSRI) [48].

There was no significant difference in the prevalence of depression in eosinophilic (25.3%) versus non-eosinophilic asthma (23.5%). This was contrary to a study done in France which found non-eosinophilic asthma to be associated with a higher rate of depression and uncontrolled asthma [49]. One possible reason for this could be that non-eosinophilic asthma is known to poorly respond to standard asthma treatments, especially to inhaled corticosteroids [50]. However, the French study only included patients with severe asthma. Currently, there is a lack of studies comparing eosinophilic and non-eosinophilic asthma across other asthma severity categories.

Those with depression had poorer quality of life compared to those without depression (AQLQ 5.53 ± 0.72 vs 6.06 ± 0.63). This was consistent with findings from previous studies [51,52]. The relation between depression and poor quality of life is likely bidirectional, as poor quality of life can be due to various stressors which can cause depression, while depression can also cause social deprivation resulting in a poor quality of life [53]. Irrespective of whether depression is a cause or effect or both feed into each other, from the patient management perspective, it is important to look for depression in those with asthma, and manage it appropriately. Furthermore, good management practices such as compliance with treatment and adopting the correct technique of using ICS, could potentially contribute to both better asthma control and improved psychological wellbeing. This study, however, is not sufficient to establish causal relationships.

## Limitations

This study was conducted in a referral hospital, which may introduce a bias towards a more severe asthma population. Thus, it may not be possible to extrapolate these results to the general Indian asthma population. There was no control group to establish a baseline prevalence of depression in the general population. Furthermore, self-reported measures are prone to be impacted by recall bias. The study was designed and powered to estimate the prevalence of depression in the cohort of Asthma patients, which was the primary outcome that was assessed. The study may not be sufficiently powered to detect differences between specific subgroups, such as eosinophilic and non-eosinophilic asthma, which were secondary outcomes.

## Conclusion

In this Indian referral cohort, nearly one in four adults with asthma screened positive for depression. Quality of life impairment emerged as the strongest independent correlate of depression, highlighting that the **subjective burden of asthma has a greater influence on psychological health than demographic factors or disease severity alone**. Female gender, presence of comorbidities, poor inhaler technique, non-adherence, and poor asthma control were also associated with higher depression rates. These findings underscore the importance of **routine screening for depression as part of comprehensive asthma management**, alongside optimization of asthma control and quality of life. Future longitudinal studies should evaluate whether interventions that improve asthma-related quality of life, or treatment of comorbid depression, can influence both psychological and respiratory outcomes.

## Supporting information

**S1 Data. Dataset used for analysis.**
(XLSX)

## Acknowledgments

We thank Mr. Srinivas S of the Department of Biostatistics, Christian Medical College, Vellore for their help in the statistical analysis.

## Author contributions

**Conceptualization:** Anushka Kataria, Jefferson Daniel, Barney Isaac, Balamugesh Thangakunam, Jesson Paulson Illimoottil, Devasahayam J Christopher.

**Formal analysis:** Anushka Kataria, Jefferson Daniel, Reka Karuppusami.

**Funding acquisition:** Devasahayam J Christopher.

**Investigation:** Anushka Kataria, Jefferson Daniel.

**Methodology:** Anushka Kataria, Jefferson Daniel, Barney Isaac, Balamugesh Thangakunam, Jesson Paulson Illimoottil, Devasahayam J Christopher.

**Project administration:** Barney Isaac.

**Supervision:** Balamugesh Thangakunam, Devasahayam J Christopher.

**Validation:** Barney Isaac.

**Writing – original draft:** Anushka Kataria.

**Writing – review & editing:** Jefferson Daniel, Barney Isaac, Balamugesh Thangakunam, Jesson Paulson Illimoottil, Reka Karuppusami, Devasahayam J Christopher.

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
