## [Decision Letter · Decision Letter 0]

19 Aug 2025

PONE-D-25-32861
Predictors of depression in asthma patients in an Indian referral hospital
PLOS ONE

Dear Dr. Daniel J,

Thank you for submitting your manuscript to PLOS ONE. After careful consideration, we feel that it has merit but does not fully meet PLOS ONE’s publication criteria as it currently stands. Therefore, we invite you to submit a revised version of the manuscript that addresses the points raised during the review process.

We look forward to receiving your revised manuscript.

Kind regards,

Bharat Bhushan Sharma, M.D.

Academic Editor

PLOS ONE

Journal Requirements:

https://link.springer.com/article/10.1007/s00213-019-05326-9?

https://pmc.ncbi.nlm.nih.gov/articles/PMC8235599

https://pubmed.ncbi.nlm.nih.gov/35772291/

In your revision ensure you cite all your sources (including your own works), and quote or rephrase any duplicated text outside the methods section. Further consideration is dependent on these concerns being addressed.

3. We note that there is identifying data in the Supporting Information file “S1_Data.xlsx”. Due to the inclusion of these potentially identifying data, we have removed this file from your file inventory. Prior to sharing human research participant data, authors should consult with an ethics committee to ensure data are shared in accordance with participant consent and all applicable local laws.

-Location data

Additional Editor Comments:

The manuscript would benefit from clarifying methodological distinctions between the depression assessment tools used, including justification for their selection and acknowledgment of potential biases inherent in self-reported measures. Statistical interpretations should more rigorously account for confounding in multivariate analyses, with clearer articulation of model adjustments and limitations. Given the study’s restricted setting, the authors should discuss implications for generalizability. Enhancing lexical variety throughout would improve clarity and impact. The study’s motivation and structure should be more explicitly stated, and the discussion strengthened by addressing comorbidities and potential links between depression and other chronic diseases.

Reviewers' comments:

Reviewer's Responses to Questions

**Comments to the Author**

1. Is the manuscript technically sound, and do the data support the conclusions?

Reviewer #1: Partly

Reviewer #2: Partly

Reviewer #3: Yes

2. Has the statistical analysis been performed appropriately and rigorously?

Reviewer #1: Yes

Reviewer #2: No

Reviewer #3: Yes

3. Have the authors made all data underlying the findings in their manuscript fully available?

Reviewer #1: Yes

Reviewer #2: Yes

Reviewer #3: Yes

4. Is the manuscript presented in an intelligible fashion and written in standard English?

Reviewer #1: No

Reviewer #2: Yes

Reviewer #3: Yes

5. Review Comments to the Author

Reviewer #1: While the study provides valuable insight into the prevalence of depression among asthma patients in India, certain aspects require clarification and cautious interpretation. The cross-sectional design precludes causal inference, yet several discussion points suggest directional relationships (e.g., improved adherence to inhaled corticosteroids leading to reduced depression) that cannot be substantiated within the present framework. Furthermore, although multiple variables were significant in univariate analyses, only the Asthma Quality of Life Questionnaire (AQLQ) score retained significance in the multivariate model, indicating that other associations may be confounded and should be interpreted accordingly. The use of a PHQ-9 cut-off score of ≥5 includes very mild cases and may overestimate prevalence compared with studies employing higher thresholds. Comparisons with prior work using different instruments, such as the Beck Depression Inventory, should more explicitly acknowledge methodological differences. The single tertiary-care setting may limit external validity, and the study may lack sufficient power to detect differences between eosinophilic and non-eosinophilic asthma. Finally, reliance on self-reported or observed measures for adherence and inhaler technique introduces potential recall and social desirability bias.

In the other hand, the manuscript is written in generally sound scientific English; however, several stylistic refinements could enhance clarity and alignment with editorial standards. Certain sentences are overly long and densely packed with numerical values or parenthetical details, which may hinder readability and would benefit from being divided or simplified. The text alternates between active and passive voice in an inconsistent manner, and a more uniform tone is advisable. Recurrent use of phrases such as “prevalence of depression” could be moderated through synonyms to improve lexical variety. In the Results and Discussion sections, occasional use of the present tense to describe study findings should be replaced with the past tense for consistency. Additionally, the insertion of appropriate punctuation—particularly commas—would aid in separating lengthy clauses and improving overall clarity.

Reviewer #2: After examining the paper, the following problems were found and the author is suggested to do revisions as mentioned below:

1) In the abstract the author is suggested to highlight the Implications by explaining the significance of these findings & also how do they contribute to the existing body of knowledge. It is also best to mention the areas where further investigation is needed based on your findings. Write a complete but concise description of your work.

2) In the introduction the author is suggested to clearly state the specific goal of this research, explaining why this problem is important and why it needs to be investigated. Also outline the paper's structure to help the reader understand the logical flow of your research and what to expect in each section.

3) Depression can be related to thyroid issues. Both hypothyroidism and hyperthyroidism can cause or worsen it. There is a potential link between asthma, depression, and thyroid issues. Studies suggest a higher prevalence of thyroid dysfunction, particularly hypothyroidism, in individuals with depression. I’ am unable to find anything regarding this in the paper. Understanding the potential link between these conditions is important for comprehensive patient care. If one has asthma and is experiencing symptoms of depression, it's crucial to assess thyroid function.

4) The conclusion shows a theoretical analysis, backed by observations and it lacks focus. The conclusion is intended to help the reader understand why your research should matter to them after they have finished reading the paper. In the conclusion the author is suggested to briefly reiterate the key points, findings and most important results of this study. Focus on the key takeaways that directly address the research and provide a sense of closure with a lasting impression.

5) The motivation of the study and the advantages of this article over those other published articles needs to be further emphasized.

Reviewer #3: Good observations and well written manuscript, expand the abbreviation at least when they came first. What were the comorbidities associated with asthma, it should be mentioned Cleary in text, whether the depression associated with comorbidities.

6. PLOS authors have the option to publish the peer review history of their article (what does this mean?). If published, this will include your full peer review and any attached files.

Reviewer #1: **Yes: **Mb. Arias

Reviewer #2: **Yes: **Dr. Abhilash

Reviewer #3: **Yes: **Dr Seema Rani

---

## [Author Response · Author response to Decision Letter 1]

10 Oct 2025

Journal Requirements:

The manuscript has been revised to meet PLOS ONE's style requirements. The dataset has been fully anonymized, with all direct identifiers and study numbers removed. We have reviewed the manuscript and rephrased the instances of overlapping text to ensure originality. All sources have been appropriately cited.

Additional Editor Comments:

1. The manuscript would benefit from clarifying methodological distinctions between the depression assessment tools used, including justification for their selection and acknowledgment of potential biases inherent in self-reported measures.

Author response: We have updated our ‘Methodology’ (last paragraph) and ‘Discussion’ (first paragraph) to clarify our choice of depression assessment tool and its comparison with other available tools. We have acknowledged the potential bias in our ‘Limitations’.

2. Statistical interpretations should more rigorously account for confounding in multivariate analyses, with clearer articulation of model adjustments and limitations.

Author response: The manuscript has been updated to include this in the ‘Results’ (last paragraph) and ‘Limitations’.

3. Given the study’s restricted setting, the authors should discuss implications for generalizability.

Author response: We acknowledge the restricted setting and have added a statement in ‘Limitations’, addressing the generalizability of our findings to populations outside similar clinical settings.

4. Enhancing lexical variety throughout would improve clarity and impact.

Author response: We have revised the manuscript to improve lexical variety.

5. The study’s motivation and structure should be more explicitly stated, and the discussion strengthened by addressing comorbidities and potential links between depression and other chronic diseases.

Author response: We have clarified the study’s motivation and structure in the final paragraph of the ‘Introduction’. Our study showed a significant association of depression with the presence of one or more comorbidities. The ‘Discussion’ (3rd paragraph) addresses findings from other research with regard to the potential link between depression and the presence of asthma, along with other comorbidities.

Reviewer #1:

While the study provides valuable insight into the prevalence of depression among asthma patients in India, certain aspects require clarification and cautious interpretation.

1. The cross-sectional design precludes causal inference, yet several discussion points suggest directional relationships (e.g., improved adherence to inhaled corticosteroids leading to reduced depression) that cannot be substantiated within the present framework.

Author response: We agree and have revised the ‘Discussion’ to avoid implying causality. In particular, the final paragraph of the ‘Discussion’ now explicitly notes that the associations observed in our study, while potentially relevant for clinical management, are not sufficient to determine causal relationships.

2. Furthermore, although multiple variables were significant in univariate analyses, only the Asthma Quality of Life Questionnaire (AQLQ) score retained significance in the multivariate model, indicating that other associations may be confounded and should be interpreted accordingly.

Author response: The manuscript has been updated to include this in the ‘Results’ (last paragraph).

3. The use of a PHQ-9 cut-off score of ≥5 includes very mild cases and may overestimate prevalence compared with studies employing higher thresholds.

Author response: Thank you for the observation. The aim of our study was to estimate the overall prevalence of depression. We have shown the breakdown of the full spectrum of severity in the ‘Results’.

4. Comparisons with prior work using different instruments, such as the Beck Depression Inventory, should more explicitly acknowledge methodological differences.

Author response: We have acknowledged this in our ‘Discussion’ (first paragraph).

5. The single tertiary-care setting may limit the external validity of the study, and it may lack sufficient power to detect differences between eosinophilic and non-eosinophilic asthma.

Author response: The study was designed and powered to estimate the prevalence of depression, which was the primary objective. Therefore, the study may not have been sufficiently powered to detect differences between specific subgroups, such as eosinophilic and non-eosinophilic asthma, which were secondary objectives. We acknowledge this in the ‘Limitations’ section. We have also added a statement addressing the study’s generalizability, given the single tertiary care setting.

6. Finally, reliance on self-reported or observed measures for adherence and inhaler technique introduces potential recall and social desirability bias.

Author response: Adherence to treatment was self-reported, and it is the most practical way of obtaining this information, despite its limitations. The inhaler technique was assessed with direct observation by an experienced investigator. Nevertheless, we have acknowledged the possibility of bias in the ’Limitations’.

7. In the other hand, the manuscript is written in generally sound scientific English; however, several stylistic refinements could enhance clarity and alignment with editorial standards. Certain sentences are overly long and densely packed with numerical values or parenthetical details, which may hinder readability and would benefit from being divided or simplified. The text alternates between active and passive voice in an inconsistent manner, and a more uniform tone is advisable. Recurrent use of phrases such as “prevalence of depression” could be moderated through synonyms to improve lexical variety. In the Results and Discussion sections, occasional use of the present tense to describe study findings should be replaced with the past tense for consistency. Additionally, the insertion of appropriate punctuation—particularly commas—would aid in separating lengthy clauses and improving overall clarity.

Author response: We have carefully revised the manuscript to enhance clarity and readability, ensuring a more consistent tone through the use of the past tense and alignment between active and passive voice, as appropriate. We have also rewritten a concise ‘Conclusion’.

Reviewer #2:

After examining the paper, the following problems were found and the author is suggested to do revisions as mentioned below:

1) In the abstract, the author is suggested to highlight the Implications by explaining the significance of these findings & also how do they contribute to the existing body of knowledge. It is also best to mention the areas where further investigation is needed based on your findings. Write a complete but concise description of your work.

Author response: The abstract has been modified to highlight the implications of the findings, while considering the 300-word limit.

2) In the introduction the author is suggested to clearly state the specific goal of this research, explaining why this problem is important and why it needs to be investigated. Also outline the paper's structure to help the reader understand the logical flow of your research and what to expect in each section.

Author response: The ‘Introduction’ has been modified to state clearly the specific aims of the research, the importance of this question, and the knowledge gaps, providing justification for this study.

3) Depression can be related to thyroid issues. Both hypothyroidism and hyperthyroidism can cause or worsen it. There is a potential link between asthma, depression, and thyroid issues. Studies suggest a higher prevalence of thyroid dysfunction, particularly hypothyroidism, in individuals with depression. I’ am unable to find anything regarding this in the paper. Understanding the potential link between these conditions is important for comprehensive patient care. If one has asthma and is experiencing symptoms of depression, it's crucial to assess thyroid function.

Author response: Thank you for the observation. Assessment of thyroid function was beyond the scope of our study, which aimed to investigate the prevalence of depression and its associated factors. Therefore, thyroid dysfunction was not evaluated, and so it does not figure in the discussion.

4) The conclusion shows a theoretical analysis, backed by observations and it lacks focus. The conclusion is intended to help the reader understand why your research should matter to them after they have finished reading the paper. In the conclusion the author is suggested to briefly reiterate the key points, findings and most important results of this study. Focus on the key takeaways that directly address the research and provide a sense of closure with a lasting impression.

Author response: Thank you for the observation. The ‘Conclusion’ has been rewritten and made concise, with improved clarity, emphasising the key findings and implications of the study.

5) The motivation of the study and the advantages of this article over those other published articles needs to be further emphasized.

Author response: The ‘Introduction’ has been modified to clarify the study’s background, and the ‘Discussion’ highlights the importance of the findings in this study.

Reviewer #3:

1. Good observations and well written manuscript, expand the abbreviation at least when they came first.

Author response: Thank you, we have revised the manuscript to ensure all abbreviations are expanded at their first mention.

2. What were the comorbidities associated with asthma, it should be mentioned Cleary in text, whether the depression associated with comorbidities.

Author response: Thank you for the observation. We have now included this in the ‘Results’.

---

## [Decision Letter · Decision Letter 1]

18 Nov 2025

Predictors of depression in asthma patients in an Indian referral hospital

PONE-D-25-32861R1

Dear Dr. Daniel J,

We’re pleased to inform you that your manuscript has been judged scientifically suitable for publication and will be formally accepted for publication once it meets all outstanding technical requirements.

Kind regards,

Bharat Bhushan Sharma, M.D.

Academic Editor

PLOS ONE

Additional Editor Comments (optional):

The authors have addressed all the queries raised during peer review, and the paper is now suitable for publication in the journal.

Reviewers' comments:

Reviewer's Responses to Questions

**Comments to the Author**

1. If the authors have adequately addressed your comments raised in a previous round of review and you feel that this manuscript is now acceptable for publication, you may indicate that here to bypass the “Comments to the Author” section, enter your conflict of interest statement in the “Confidential to Editor” section, and submit your "Accept" recommendation.

Reviewer #3: All comments have been addressed

2. Is the manuscript technically sound, and do the data support the conclusions?

Reviewer #3: Yes

3. Has the statistical analysis been performed appropriately and rigorously?

Reviewer #3: Yes

4. Have the authors made all data underlying the findings in their manuscript fully available?

Reviewer #3: Yes

5. Is the manuscript presented in an intelligible fashion and written in standard English?

Reviewer #3: Yes

6. Review Comments to the Author

Reviewer #3: All queries have been answered. Manuscript has been well rewritten. No further correction/comments needed from our side.

7. PLOS authors have the option to publish the peer review history of their article (what does this mean?). If published, this will include your full peer review and any attached files.

Reviewer #3: No

---

## [Editor Report · Acceptance letter]

PONE-D-25-32861R1

PLOS One

Dear Dr. Daniel,

I'm pleased to inform you that your manuscript has been deemed suitable for publication in PLOS One. Congratulations! Your manuscript is now being handed over to our production team.

Kind regards,

on behalf of

Professor Bharat Bhushan Sharma

Academic Editor

PLOS One